# Cytotoxic Potential of the Monoterpene Isoespintanol against Human Tumor Cell Lines

**DOI:** 10.3390/ijms25094614

**Published:** 2024-04-23

**Authors:** Orfa Inés Contreras-Martínez, Alberto Angulo-Ortíz, Gilmar Santafé Patiño, Fillipe Vieira Rocha, Karine Zanotti, Dario Batista Fortaleza, Tamara Teixeira, Jesus Sierra Martinez

**Affiliations:** 1Biology Department, Faculty of Basic Sciences, University of Córdoba, Montería 230002, Colombia; oicontreras@correo.unicordoba.edu.co; 2Chemistry Department, Faculty of Basic Sciences, University of Córdoba, Montería 230002, Colombia; gsantafe@correo.unicordoba.edu.co; 3Chemistry Department, Federal University of São Carlos, São Carlos 13565-905, Brazil; fillipe@ufscar.br (F.V.R.); karinezanotti_@hotmail.com (K.Z.); tamara.teixeira.296@gmail.com (T.T.); 4Genetics and Evolution Department, Federal University of São Carlos, São Carlos 13565-905, Brazil

**Keywords:** antitumor activity, isoespintanol, monoterpene, cell lines

## Abstract

Cancer is a disease that encompasses multiple and different malignant conditions and is among the leading causes of death in the world. Therefore, the search for new pharmacotherapeutic options and potential candidates that can be used as treatments or adjuvants to control this disease is urgent. Natural products, especially those obtained from plants, have played an important role as a source of specialized metabolites with recognized pharmacological properties against cancer, therefore, they are an excellent alternative to be used. The objective of this research was to evaluate the action of the monoterpene isoespintanol (ISO) against the human tumor cell lines MDA-MB-231, A549, DU145, A2780, A2780-cis and the non-tumor line MRC-5. Experiments with 3-(4,5-dimethylthiazol-2-yl)-2,5-diphenyltetrazolium bromide (MTT) and fluorescence with propidium iodide (PI), 4′,6-diamidino-2-phenylindole dilactate (DAPI) and green plasma revealed the cytotoxicity of ISO against these cells; furthermore, morphological and chromogenic studies revealed the action of ISO on cell morphology and the inhibitory capacity on reproductive viability to form colonies in MDA-MB-231 cells. Likewise, 3D experiments validated the damage in these cells caused by this monoterpene. These results serve as a basis for progress in studies of the mechanisms of action of these compounds and the development of derivatives or synthetic analogues with a better antitumor profile.

## 1. Introduction

Non-communicable diseases (NCDs), also known as chronic diseases, are medical conditions that are associated with long duration and slow progress; currently, they are responsible for the majority of deaths globally. Cancer is one of the four main NCDs with the highest number of deaths [1,2]. In 2020, almost 10 million deaths were attributed to this disease [3,4]. It is an illness that affects any part of the body, presenting typical characteristics such as abnormal cells, with growth and potential to spread to different parts of the body. According to estimates by the World Health Organization (WHO) in 2015, cancer is the first or second cause of death before the age of 70 in 91 out of 172 countries and ranks third or fourth in 22 other countries. The incidence of this disease has increased rapidly in recent years [5,6]. The most common cancers in 2020, in terms of new cases, included breast (2.26 million cases); lung (2.21 million cases); colorectal (1.93 million cases); prostate (1.41 million cases); skin, other than melanoma (1.20 million cases); and gastric (1.09 million cases) [4].

Despite the existence of various methods to treat cancer, there are limitations inherent to each method and also to access the affected part. Chemotherapy drugs have been used for years, but serious side effects in patients have limited their use [7]. Furthermore, resistance to chemotherapy is a major obstacle to the effective treatment of this disease [8]. In this context, the search and development of novel, chemopreventive and antitumor compounds that act against tumor cells with minimal effect on healthy cells is urgent. Various investigations with natural or synthetic compounds have documented their ability to inhibit the growth of tumor cells both in vitro and in vivo; many of these compounds have antioxidant, antiproliferative and pro-apoptotic effects in several types of cancer [9,10,11,12].

Monoterpenes are a group of natural products that have been widely studied for their therapeutic potential against various pathologies [13]. They have been the research subject in the discovery of antineoplastic drugs, since they have a potential antitumor effect and low toxicity. Furthermore, the structural diversity of these compounds provides greater plasticity when interacting with these cells [7,14,15,16]. Monoterpenes such as carvacrol and linalool have shown tumor inhibition both in vitro and in vivo in various types of cancers, revealing diverse mechanisms of action that include induction of apoptosis, impairment of the cell cycle, production of reactive oxygen species (ROS), autophagy and necroptosis among others [13]. Furthermore, carvacrol and thymol have shown antitumor and antiproliferative activity through several signaling pathways [17]. Likewise, the anticancer property of D-limonene has also been reported in several types of cancer, indicating that it could be due to its ability to trigger apoptosis and regulate the cell cycle [18,19]. Jointly, it has been documented that citral could be a potential new agent that can eliminate drug-resistant breast cancer cells by inducing apoptosis [14]. Also, the monoterpenes nerol, neral, geranial and geraniol have shown promising anticancer activity that has inspired new analogous syntheses [20]. 

Isoespintanol (ISO) [2-isopropyl-3,6-dimethoxy-5-methylphenol] is a monoterpene first obtained from the aerial parts of *Eupatorium saltense* (Asteraceae) [21], and its synthesis was previously reported [22]. Additionally, it has been extracted from *Oxandra xylopioides* (Annonaceae). Chemical and biological activities, such as antioxidant [23], anti-inflammatory [24], antispasmodic [25], vasodilator [26] and cryoprotective of canine semen, have been reported for this compound [27], as well as insecticidal [28] and antifungal activities against *Colletotrichum* [29]. Furthermore, its activity against human pathogenic bacteria and yeasts has also been documented [30,31,32,33]. The cytotoxicity of this compound has been studied in the line of immortalized African green monkey kidney epithelial cells (VERO) [32], human peripheral blood lymphocytes [34] and murine macrophages (RAW 264.7) [35]. However, there are no reports on its potential against tumor cells; therefore, we hypothesize that ISO has a cytotoxic effect against tumor cells based on the wide range of described activities. Considering the growing increase in the incidence of different types of cancer in recent years, which include among the most common breast, lung and prostate cancer, the purpose of this research was to evaluate the antitumor activity of ISO against lung (A549), breast (MDA-MB-231), prostate (DU145) and ovarian (A2780 and A2780-cis) tumor cell lines, contributing to the search for new compounds of natural origin that can serve as adjuvants in the treatment of cancer.

## 2. Results

### 2.1. MTT Assay

The cytotoxic activity of isoespintanol (ISO) was evaluated against different human tumor cell lines (MDA-MB-231, A549, DU145, A2780 and A2780-cis) and the non-tumor line (MRC-5) using the MTT assay. From the data obtained, the IC_50_ values (inhibitory concentration of 50% of cell viability) of the ISO were calculated for all the cell lines under study (Table 1). The ISO showed cytotoxicity against all the cell lines evaluated. IC_50_ values of cisplatin on MDA-MB-231, A549, DU145, A2780, A2780-cis and MRC-5 monolayers are shown in Appendix A.

### 2.2. Cell Morphology Assay

The assay to evaluate the effect of ISO on the morphology of the MDA-MB-231 and MRC-5 cell lines revealed morphological changes in both lines. In cells treated at concentrations of 0.5IC_50_ (26.2 µM and 19.97 µM for MDA-MB-231 and MRC-5, respectively), no significant changes were observed, while at concentrations of IC_50_ (52.4 and 39.95 µM, respectively) and 2IC_50_ (104.8 and 79.9 µM, respectively) after 24 h of treatment, important morphological changes were observed such as nuclear and cytoplasmic condensation, decrease in cell confluency and cell fragmentation surrounded by membranes (smaller cells); these changes were observed in individual cells, which is characteristic of death by apoptosis [36]. The tests with propidium iodide (PI) showed cells stained red, evidencing cell death, which grew as the IC_50_ values increased, so we proposed that ISO induced cell death by apoptosis in both cell types (Figure 1 and Figure 2).

The morphological changes in the MDA-MB-231 cells treated with ISO were also evidenced with fluorescence studies using green plasma and DAPI, as seen in Figure 3. The increase in the ISO concentration (IC_50_ and 2IC_50_) caused a greater effect on the morphology of these cells.

### 2.3. Clonogenicity Assay

MDA-MB-231 breast tumor cells and the non-tumor lung cell line MRC-5 were treated with ISO (0.5IC_50_, IC_50_ and 2IC_50_) for 48 h to evaluate their ability to survive, reproduce and form colonies after the treatment with this monoterpene. The cells were evaluated after a period of 10 days. The number of colonies of MDA-MB-231 cells treated with ISO decreased significantly. In contrast, the number of colonies in MRC-5 cells treated with the ISO only decreased significantly at the concentration of 2IC_50_ (Figure 4). The antiproliferative effect of ISO was higher in MDA-MB-231 cells compared to MRC-5 cells, demonstrating a dose-dependent effect.

### 2.4. Three-Dimensional Assays

The 3D cell culture model offers advantages that are not possible in the traditional 2D model, bringing it closer to the in vivo model. In 3D culture, cells are organized in a three-dimensional way that favors cell–cell and cell–matrix interactions. The 3D tumor cell environment features a differentiated proliferation rate, oxygen gradient, nutrient gradient, acidic pH, the presence of apoptotic and necrotic cell death, regulation of gene expression and drug resistance. While in the 2D model, the cells are forced to adapt to a rigid surface, the distribution of nutrients and oxygen occurs uniformly, there is less cell–cell interaction and dead cells are removed [36,37,38,39,40].

The experiment tested the ISO at five different concentrations: 0.5, 1, 2, 4 and 8 times the IC_50_ value determined in the 2D experiment against the MDA-MB-231 cell line. The spheroid cultured in the standard medium served as the control. The spheroid control maintained its cell density and spherical shape throughout the analyzed periods. In contrast, the spheroid exposed to the ISO exhibited cell detachment after 8 days. Notably, 2IC_50_, 4IC_50_ and 8IC_50_ concentrations highlighted the expected trend, with spheroids displaying greater resistance than monolayer cells. The most significant impact was observed at higher concentrations. After 10 days, the spheroid dissolved, accompanied by a noticeable decrease in cell density at 4IC_50_ and 8IC_50_ concentrations, indicating the compound’s efficacy, as shown in Figure 5. This observation was further validated by images using DAPI and PI markers, confirming the compound’s action. Only live cells were observed in the control group after 10 days, with DAPI marking the cells. However, treatment with the compound at various concentrations resulted in cell death, as indicated by a PI marker (Figure 6).

## 3. Discussion

The great cellular heterogeneity in the tumor microenvironment and the astonishing ability of cancer cells to mutate, and thus evade almost any therapeutic intervention, support the lethality of the disease and have proven to be immune to many decades of efforts to overcome it [41]. Cancer drugs account for approximately a quarter of total new drug discovery and approvals each year. Although some of the treatments used help some patients, the great challenges faced in cancer make other therapeutic strategies necessary to reduce the burden of the disease [42]. Therefore, the search and development of new compounds with antitumor potential, that are effective and safe, as well as the development of new treatment strategies with better patient tolerance are urgent today. Since ancient times, natural products have made an important contribution to pharmacotherapy, especially in cancer and infectious diseases. In this scenario, plants play a primary role as a source of specialized metabolites with recognized medicinal properties [43,44]. Due to their wide chemical diversity, these metabolites can be used directly as bioactive compounds, as drug prototypes or used as pharmacological tools for different targets [45].

In this research, we demonstrate that the monoterpene ISO extracted from *O. xylopioides* inhibits the development of the tumor cell lines MDA-MB-231, A549, DU145, A2780-cis, A2780 and the non-tumor line MRC-5 with IC_50_ values of 52.39 µM, 59.70 µM, 47.84 µM, 60.35 µM, 42.15 µM and 39.95 µM, respectively. Monoterpenes have been well documented for their antitumor activity and chemopreventive activity against various types of cancer. Among the mechanisms of action that may explain the antitumor activity of these compounds is the induction of phase II carcinogen-metabolizing enzymes, resulting in carcinogenic detoxification. Considering this, the tumor suppressive chemopreventive activity of these compounds may be due to the induction of apoptosis and/or the inhibition of post-translational isoprenylation of cell growth regulatory proteins and transforming growth factor 1; therefore, monoterpenes seem to act through multiple mechanisms in cancer chemoprevention and chemotherapy [46].

ISO caused changes in the morphology of MDA-MB-231 cells (Figure 1, Figure 2 and Figure 3), such as compaction of the cell nucleus and cytoplasm and cell fragments surrounded by membrane, which is indicative of cell death by apoptosis. These results are consistent with studies reported with citral, which can eliminate drug-resistant breast cancer cells (MDA-MB-231) in a spheroid model by inducing apoptosis [14] and inhibit the growth of MCF-7 breast cells with an arrest of the cell cycle in the G2/M phase and induction of apoptosis. Considering this, a decrease in the synthesis of prostaglandin E2 was observed, which supports a possible chemopreventive effect of this compound [44].

Other monoterpenes, such as limonene, have been documented with chemopreventive activity, stimulating the detoxification of carcinogenic compounds and limiting tumor growth and angiogenesis in various cancer models. The anticancer activity of limonene was related to the inhibition of tumor initiation, growth and angiogenesis and the induction of cancer cell apoptosis [18,19]. The activity of geraniol has also been documented against cancer of the prostate, intestine, liver, kidney and skin due to the induction of apoptosis and the increase in the expression of pro-apoptotic proteins. Furthermore, it has been indicated that the synergy of this compound with other drugs can further increase the range of chemotherapeutic agents [47]. The monoterpenoids nerol, geranial and neral have been reported with antineoplastic activity in animal and cellular models in various types of cancer; these compounds have been found to activate multiple antitumor responses, such as apoptosis, autophagic cell death, cytostasis and necrosis. These multitarget mechanisms may provide superior therapeutic effects, reducing the adaptive resistance of tumor cells [20].

We also evaluated the ability of MDA-MB-231 and MRC-5 cells to survive and reproduce to form colonies after ISO treatment. The results show the capacity of ISO to inhibit colony formation in both cell types at concentrations of IC_50_ and 2IC_50_, with the effect observed on the MDA-MB-231 tumor cell line being significantly greater, compared to the effect observed on the non-tumor cell line MRC-5. These results are consistent with those documented with other monoterpenes, for which antiproliferative capacity has been demonstrated. Thymoquinone has shown its effectiveness in inhibiting different stages of cancer, such as proliferation, migration and invasion. In addition, it acts as an anticancer agent against different human cancers by inducing apoptosis and regulating levels of pro- and anti-apoptotic genes and inhibition of metastasis [48]. Furthermore, monoterpenes such as iridoid glycosides also exert inhibitory effects on numerous cancers; these compounds inhibit cancer growth by inducing cell cycle arrest or by regulating signaling pathways related to apoptosis. In addition, they suppress the expression and activity of matrix metalloproteinases (MMPs), which reduces the migration and invasiveness of cancer cells. The antiangiogenic mechanism of iridoid glycosides was found to be closely related to the transcriptional regulation of proangiogenic factors, i.e., vascular endothelial growth factors (VEGFs) and cluster of differentiation 31 (CD31). In this way, these compounds can alleviate or prevent the rapid progression of cancer and metastasis [49].

Other terpenes and their derivatives have demonstrated significant anticancer potential. The sesquiterpenes β-caryophyllene (BCP) and β-caryophyllene oxide (BCPO) have important anticancer activities, affecting the growth and proliferation of numerous cancer cells, properties that have been related to their chemical structure [50]. Thus, BCPO contains exocyclic methylene and epoxide functional groups, which is why it binds covalently to proteins and DNA bases through sulfhydryl and amino groups. For this reason, BCPO reveals a high potential to be a signaling modulator in cancerous tumor cells. The anticancer activities of both sesquiterpenes can be exerted by suppressing cell growth and inducing apoptosis [51]. The 2-cyano-3,12-dioxooleana-1,9(11)-dien-28-oic acid (CDDO) and its C28 modified derivative methyl-ester (CDDO-Me, also known as bardoxolone methyl) are two synthetic derivatives of oleanolic acid (natural triterpene). These molecules have been widely investigated for their great capacity to exert antiproliferative, antiangiogenic and antimetastatic activities and to induce apoptosis and differentiation in cancer cells. At doses greater than 100 nM, CDDO and CDDO-Me are capable of modulating the differentiation of a variety of cell types, both tumor cell lines and primary culture cells. Meanwhile, at micromolar doses, these compounds exert an anticancer effect of multiple ways by inducing extrinsic or intrinsic apoptotic pathways or autophagic cell death, by inhibiting telomerase activity, altering mitochondrial functions through inhibition of the Lon protease and blocking the deubiquitylating enzyme USP7 [52]. We also validated the effectiveness of ISO against MDA-MB-231 cells through 3D experiments, using images that used DAPI and PI markers, being the first report of this type of assay in the ratification of cell damage by this compound.

The results of this research highlight the effect of ISO against the human tumor cell lines MDA-MB-231, A549, DU145, A2780, A2780-cis and the non-tumor line MRC-5, suggesting as possible mechanisms of action on MDA-MB-231 cells the induction of apoptosis and inhibition of cell proliferation. Additional studies are necessary to delve into the mechanisms of antitumor action of this monoterpene as a possible adjuvant in the treatment of this disease.

## 4. Materials and Methods

### 4.1. Obtaining and Identification of Isoespintanol

ISO was obtained as a crystalline amorphous solid from the petroleum benzine extract of *O. xylopioides* leaves, and its structural identification was performed by GC–MS, ^1^H-NMR, ^13^C-NMR, DEPT, COSY ^1^H-^1^H, HMQC and HMBC. Information related to obtaining and identifying the ISO was reported in our previous study [33].

### 4.2. Cell Lines and Reagents

The cell lines MDA-MB-231 (triple-negative human breast adenocarcinoma of mesenchymal phenotype, ATCC No. HTB-26); A549 (alveolar basal cell epithelial adenocarcinoma of the human lung, ATCC No. CCL-185); DU145 (prostate tumor, ATCC No. HTB-81); A2780-cis; (cisplatin-resistant human ovarian tumor, ECACC No. 93112517); A2780 (human ovarian tumor, ECACC No. 93112519); and MRC-5 (human non-lung tumor, ATCC No. CCL-171) were obtained from the American Type Culture Collection (ATCC) (Rockville, MD, USA). Dimethylsulfoxide (DMSO), trypan blue and (3-(4,5-dimethylthiazol-2-yl)-2,5-diphenyltetrazolium bromide) [MTT] were obtained from Sigma-Aldrich^®^ (St. Louis, MO, USA); crystal violet (CV) was obtained from Synth (Diadema, Sao Paulo, Brazil); propidium iodide was obtained from BD Biosciences (San Jose, CA, USA); trypsin and fetal bovine serum (FBS) were obtained from Vitrocell (Campinas, Sao Paulo, Brazil); phosphate-buffered saline (PBS) was prepared with sodium chloride, potassium chloride, disodium phosphate and monopotassium phosphate obtained from Synth (Diadema, Sao Paulo, Brazil); Dulbecco’s Modified Eagle culture medium (DMEM) was prepared with DMEM (D5523), gentamicin and amphotericin B obtained from Sigma-Aldrich^®^ (St. Louis, MO, USA); penicillin was obtained from Vitrocell (Campinas, Sao Paulo, Brazil); sodium bicarbonate was obtained from Neon (Suzano, Sao Paulo, Brazil); RPMI medium was prepared with RPMI-1640 (R6504), gentamicin and amphotericin B obtained from Sigma-Aldrich^®^ (St. Louis, MO, USA); penicillin was obtained from Vitrocell (Campinas, Sao Paulo, Brazil); and green plasma, CellMask™ and DAPI (4′,6-Diamidino-2-phenylindole dilactate) were obtained from Invitrogen by Thermo Fisher Scientific (Eugene, OR, USA).

### 4.3. Cell Cultures

The A2780-cis and A2780 cell lines were cultured in RPMI 1640 medium supplemented with 10% fetal bovine serum (FBS) and 1% penicillin/streptomycin, and the MDA-MB-231, A549, DU145 and MRC-5 lines were cultured in DMEM medium and were maintained in a humidified atmosphere with 5% CO_2_ at 37 °C. Medium changes were performed every 2 to 3 days for maintenance. Subcultures were carried out twice a week until reaching approximately 80% confluency, using an inverted microscope, Nikon Eclipse TS 100 (Nikon, Melville, NY, USA) for the assays.

### 4.4. MTT Assay

The cytotoxic activity of ISO against human tumor cell lines was evaluated by the colorimetric MTT assay, which is based on the ability of dehydrogenase enzymes of metabolically viable cells to reduce tetrazolium rings and form formazan crystals; consequently, the number of viable cells is directly proportional to the level of formazan produced [53,54,55]. Cisplatin was used as a reference compound [8,14] with all the cells used in this study (the IC_50_ values obtained are shown in Appendix A). This considered, the resistant commercial cell line A2780-cis with resistance to cisplatin was used. For the experiments, a density of 1.5 × 10^4^ cells/well in 150 µL of RPMI 1640 medium (for A2780 and A2780-cis) and DMEM (for MDA-MB-231, A549, DU145 and MRC-5) was seeded in 96-well microplates (Kasvi, São José dos Pinhais, Brazil), allowing their adhesion and proliferation during 24 h of incubation at 37 °C with 5% CO_2_. After this time, 0.75 µL of the ISO stock solution dissolved in DMSO at different concentrations (4.88 to 625 µM) was added to the final reaction wells, and the plates were incubated for 48 h at 37 °C. After incubation, 50 µL of MTT (1 mg/mL) was added to each well. The plates were incubated with 5% CO_2_ at 37 °C for 4 h. Subsequently, the MTT was discarded, and the plates were allowed to dry at room temperature; then, the formazan crystals deposited at the bottom of each well were dissolved in 100 µL of isopropanol. Absorbance was determined at an optical density (OD) of 540 nm using an Epoch 2 microplate reader (Biotek, Winooski, VT, USA). Cells with DMSO were used as controls. The experiments were performed three times in triplicate. The results of the tests were expressed by dose-response curves using 8 concentrations of ISO previously described. IC_50_ values (50% inhibitory concentration of the cell population) were calculated from the fit (R^2^ > 0.95) of the Hill slope curve of the experimental data using nonlinear regression analysis in GraphPad Prism version 8.0 software.

### 4.5. Cell Morphology Assay

The effect of ISO on the morphology of the MDA-MB-231 cell line was evaluated. For the assay, 1500 µL of a cell suspension was added to 12-well dishes (1 × 10^5^ cells/well) and incubated for 24 h at 37 °C and 5% CO_2_. Then, the cells were treated with 7.5 µL of the ISO (0.5IC_50_, IC_50_ and 2IC_50_) and incubated for 48 h at 37 °C with 5% CO_2_. Subsequently, the cells were observed and photographed at 0 h and 24 h. Then, the medium was removed from the cells, and they were fixed with 1 mL of methanol for 10 min. Next, the methanol was removed, 500 µL of PBS and 200 µL of green plasma were added to the wells and the plates were incubated for 30 min. Subsequently, the green plasma was removed, 500 µL of PBS and 200 µL of DAPI were added and the plates were incubated again for 5 min. Staining with propidium iodide (PI) [100 µL] for 10 min was also performed. Then, the cells were photographed with the help of the digital image capture system CELENA^®^ S Digital Imaging System (Logos Biosystems, Annandale, VA, USA). The morphology of cells treated with ISO was compared with the morphology of cells treated with 0.5% DMSO, used as a negative control. The experiments were carried out in triplicate.

### 4.6. Clonogenicity Assay

To evaluate the reproductive viability of MDA-MB-231 cells after treatment with ISO, a clonogenicity test [50] was performed. The criterion used was the number of colonies. A total of 1500 µL of the cell suspension was seeded in 6-well plates (1000 cells/well) and incubated at 37 °C with 5% CO_2_ for 24 h. Then, the cells were treated with 10 µL of ISO (0.5IC_50_, IC_50_ and 2IC_50_) and incubated for 48 h. Subsequently, the medium was discarded from the dishes, the cells were washed with 2 mL of PBS and 2 mL of DMEM culture medium supplemented with 10% FBS added, and the dishes were incubated (5% CO_2_ at 37 °C) for 10 days. Finally, the supernatant was discarded, and the cells were washed with PBS and fixed with a solution of methanol (2 mL) and acetic acid (3:1 (*v*/*v*)) for 5 min. Colonies were stained with crystal violet (0.05% crystal violet, 1% formaldehyde, 1X PBS and 1% methanol). Colony counting was performed with ImageJ 1.53t (Wayne Rasband and contributors NIH, USA), Java 1.8.0_345 image analysis software [56], and Prisma software version 8.0 was used. In determining the plating efficiency [PE] (which allows knowing the number of colonies formed based on the cells that were seeded) and the survival fraction [SF] (which allows knowing the number of colonies developed after applying the treatment with ISO) [57], to calculate the PE and the SF, the following formulas were used [58]:PE = No. of colonies formed/No. of seeded cells × 100%
SF = No. of colonies formed after treatment/No. of seeded cells × PE

### 4.7. Three-Dimensional Assays

For the 3D cell culture assay, the 96-well Bioprint Kit from Magnetic 3D Cell Culture Technology m3D-Greiner (Greiner Bio-One, Frickenhausen, Germany) was used. First, 150 µL of a solution containing nanoparticles (NanoShuttle—PL) was added to a cell culture flask containing the MDA-MB-231 cell line. After 24 h, the nanoparticle cells were plated in a 96-well repellent culture plate (3000 cells/well). The culture plate was placed under a magnetic drive to form spheroids. The plate was kept in an incubator (37 °C 5% CO_2_), and the formation and growth of the spheroids were monitored using a CELENA^®^ S Digital Imaging System microscope (Logos Biosystems). After 4 days, different concentrations of the compound were added to the spheroids, and the experiment was monitored for 10 days, with 50% of the culture medium being replaced every 48 h during the treatment period. The fluorescent markers DAPI and PI were added on the last day of treatment.

### 4.8. Data Analysis

Results were analyzed using GraphPad Prism version 8.0 software and ImageJ 1.53t (Wayne Rasband and contributors NIH, USA), Java 1.8.0_345 image analysis software.

## 5. Conclusions

This research evaluated the activity of ISO against five human tumor cell lines and one non-tumor line. The results show that ISO has a cytotoxic effect against the cell lines studied, affecting cell morphology and interrupting their reproductive viability. Thus, ISO becomes a compound of interest to continue investigating the mechanisms of action against tumor cells. Furthermore, it could be used as a natural template for the synthesis or semi-synthesis of analogues and thus could contribute to the search for alternatives that help in the control of challenging diseases such as cancer.

## Figures and Tables

**Figure 1 ijms-25-04614-f001:**
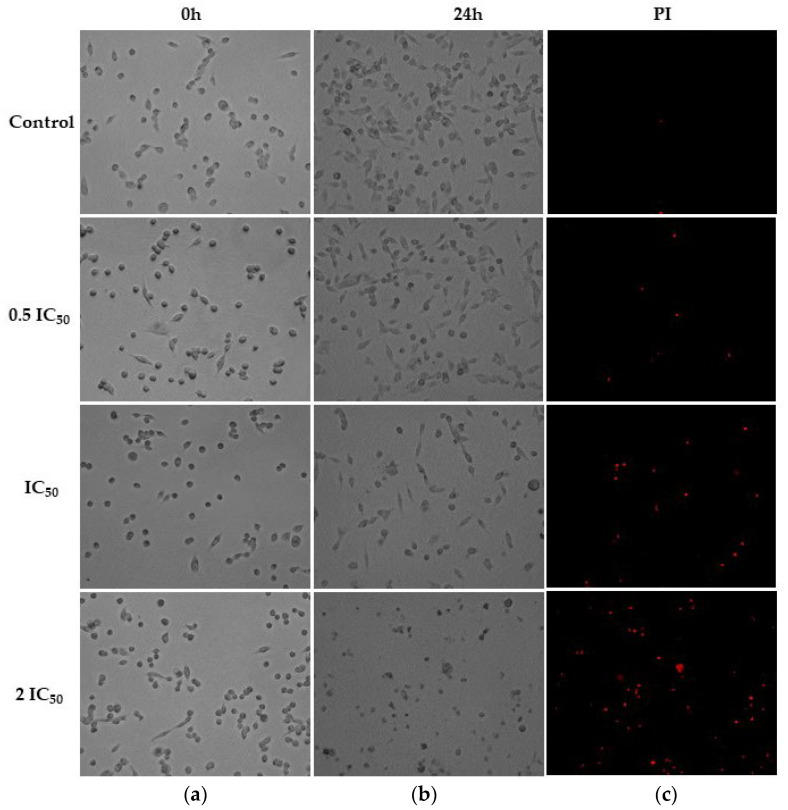
Bright-field microscopy (**a**,**b**) and fluorescence with propidium iodide (**c**) of morphological study in breast tumor cells (MDA-MB-231) after treatment with ISO (0.5IC_50_, IC_50_ and 2IC_50_) at 0 and 24 h; 10× magnification. Significant morphological changes are observed with the IC_50_ and 2IC_50_ treatments at 24 h. The cells were photographed with the CELENA^®^ S Digital Imaging System (Logos Biosystems).

**Figure 2 ijms-25-04614-f002:**
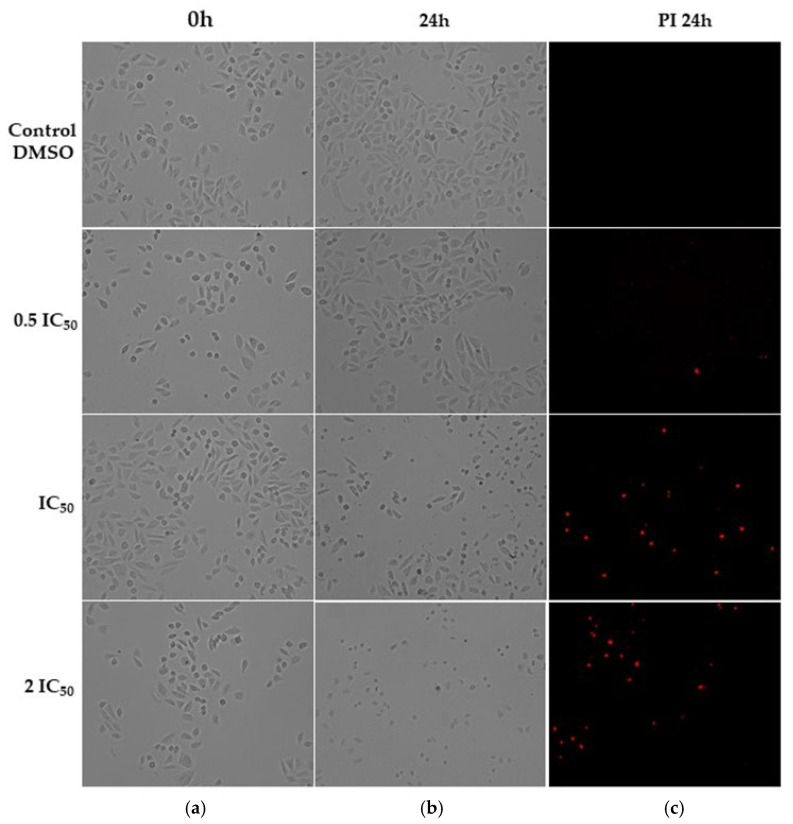
Bright-field microscopy (**a**,**b**) and fluorescence with propidium iodide (**c**) of morphological study in non-tumor lung cells (MRC-5) after treatment with ISO (0.5IC_50_, IC_50_ and 2IC_50_) at 0 and 24 h; 10× magnification. Morphological changes are observed with IC_50_ and 2IC_50_ treatments at 24 h. The cells were photographed with the CELENA^®^ S Digital Imaging System (Logos Biosystems).

**Figure 3 ijms-25-04614-f003:**
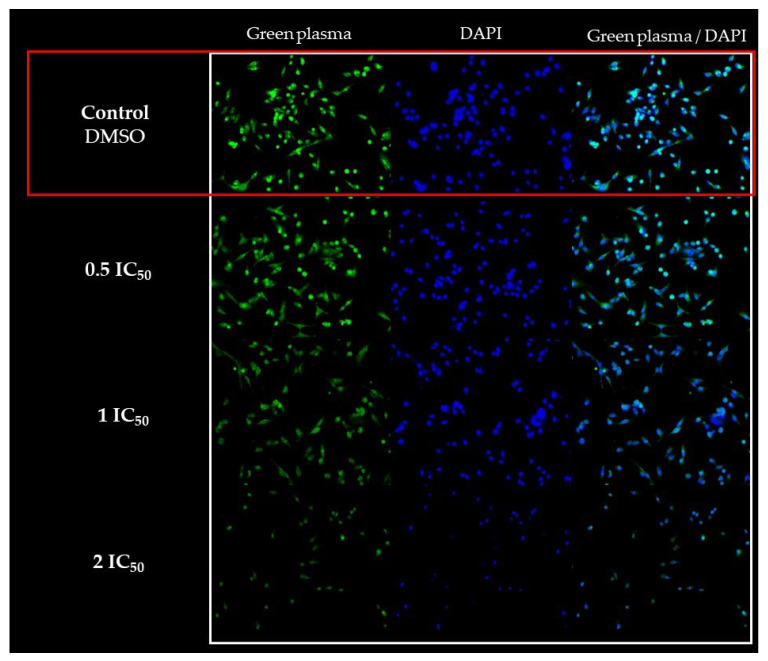
Morphological changes in breast tumor cells (MDA-MB-231) after treatment with ISO (0.5IC_50_, IC_50_ and 2IC_50_) for 24 h; 10× magnification. Double fluorescent staining (green plasma/DAPI). The cells were photographed with the CELENA^®^ S Digital Imaging System (Logos Biosystems).

**Figure 4 ijms-25-04614-f004:**
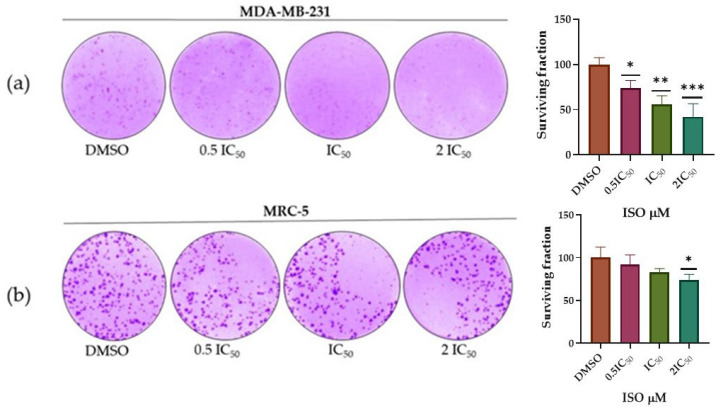
Colony formation by the breast tumor cell line MDA-MB-231 and the non-tumor lung cell line MRC-5 after plating and treatment with ISO (0.5IC_50_, IC_50_ and 2IC_50_) after 10 days of incubation; 1.1× magnification. In (**a**), the results of the Dunnett’s test with a 95% confidence level showed values of * *p* = 0.037 (DMSO vs. 0.5IC_50_), ** *p* = 0.002 (DMSO vs. IC_50_) and *** *p* = 0.0004 (DMSO vs. 2IC_50_), indicating that there are statistically significant differences between MDA-MB-231 cells treated with different concentrations of ISO and the control group. In (**b**), with the MRC-5 cells, the results of the Dunnett’s test with a confidence level of 95% showed values of * *p* = 0.020, statistically significant only with the highest ISO concentration evaluated (DMSO vs. 2IC_50_).

**Figure 5 ijms-25-04614-f005:**
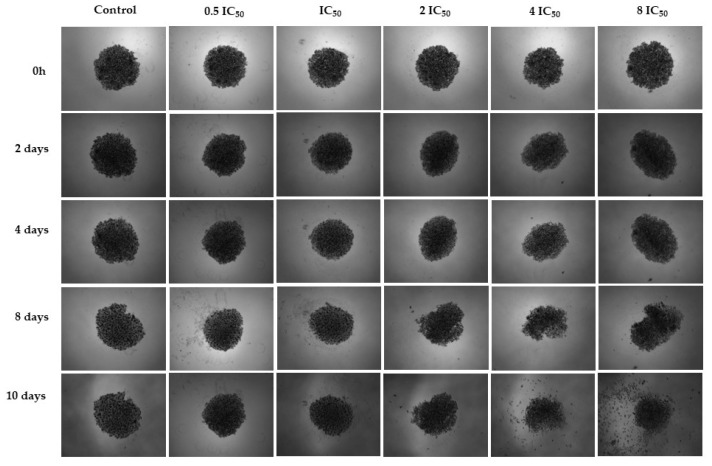
Bright-field microscopy of MDA-MB-231 cell spheroids treated with ISO (2IC_50_, 4IC_50_ and 8IC_50_) and without treatment; 4× magnification. Cell detachment was evident after 8 days of treatment with ISO.

**Figure 6 ijms-25-04614-f006:**
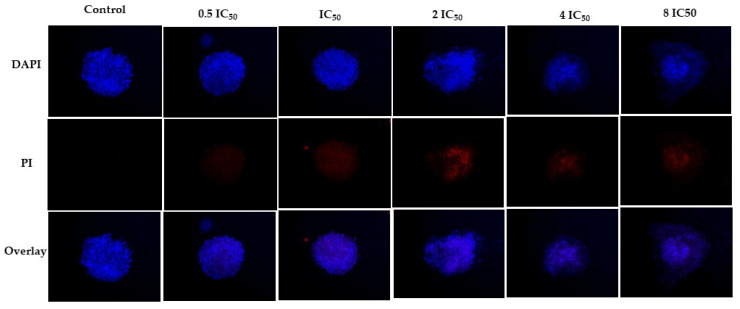
Fluorescence microscopy of MDA-MB-231 cell spheroids, labeled with DAPI and PI, treated with ISO (2IC_50_, 4IC_50_ and 8IC_50_) and without treatment, 4× magnification. Cell detachment was evident after 8 days of treatment with ISO. PI staining shows cell death at all concentrations evaluated.

**Table 1 ijms-25-04614-t001:** IC_50_ values of ISO on MDA-MB-231, A549, DU145, A2780, A2780-cis and MRC-5 monolayers.

Cell Lines	IC_50_ ISO (µM)
MDA-MB-231	52.39 ± 3.20
A549	59.70 ± 2.74
DU145	47.84 ± 3.52
A2780-cis	60.35 ± 8.4
A2780	42.15 ± 1.39
MRC-5	39.95 ± 3.76

## Data Availability

The data presented in this study are available in the article.

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
