# Peer review of "Cytotoxic Potential of the Monoterpene Isoespintanol against Human Tumor Cell Lines"

_ijms, 2024, doi:10.3390/ijms25094614_

Round 1
Reviewer 1 Report
Comments and Suggestions for Authors
New attempts on the use of natural compounds on human tumor cell lines are ceartinly welcome. The topic is big essential and the results are looking promising and deserves publication in the journal. I would recommend its publication. However, there are several issue that the authors should address before publication as follows;
1) Lines 53-56
...Monoterpenes are a group of natural products that have been widely studied for their therapeutic potential against various pathologies [13]. They have been the research subject in the discovery of antineoplastic drugs, since they have a potential antitumor effect and low toxicity. Furthermore, the structural diversity of these compounds provides greater plasticity when interacting with these cells [7,14–16]...
More details on therapeutic potential, including recent refs. on the studies of monoterpens should be given.
2) Lines 68-72
...However, there are no reports on its potential against tumor cells, then we 68 hypothesize that ISO has a cytotoxic effect against them based on the wide range of described activities. The purpose of this research was to evaluate the antitumor activity of ISO against lung (A549), breast (MDA-MB-231), prostate (DU145) and ovarian (A2780 and A2780-cis) tumor cell lines, contributing to the search for new compounds of natural origin that can serve as adjuvants in the treatment of cancer...
It is not clear why the authors select this cell line?
3) The used both trypsin and fetal bovine serum (FBS) model is not common.
are those produce in Brazil? or more detail should be given.
4) The authors try the other reagent instead of trypsin?, which may effect the results.
Comments on the Quality of English Languageminor revision
Reviewer 2 Report
Comments and Suggestions for Authors
-
-
-
- 1. The authors claim a significant decrease in the number of colonies in MDA-MB-231 cells compared to MRC-5 cells; however, the lack of statistical significance calculation or graphical representation undermines the credibility of this assertion.
-
- 2. The mention of "2.4.3 and 4.7. D Assay" appears to be a typographical error, presumably meant to be "3D assays".
-
- 3. The study's MTT assay lacks appropriate references to validate the resistance of the cell lines under investigation. The absence of such references could lead to questions about the assay's relevance and the interpretability of its results.
-
- 4. The authors discuss the potential mechanisms of action of the compounds, including the induction of phase II carcinogen-metabolizing enzymes leading to carcinogenic detoxification, but fail to provide references.
-
- 5. The paper states that cells were cultured in "DMEN medium", which likely is a typographical error for "DMEM" (Dulbecco's Modified Eagle Medium).
-
- 6. The authors' assertion that ISO is a promising compound for further investigation into anticancer mechanisms is critiqued as overly optimistic. The compound's high toxicity to healthy fibroblasts, coupled with its efficacy at high concentrations, raises concerns. The conclusion that ISO is promising is deemed overstated, especially without sufficient evidence to support its selectivity or efficacy at lower, more therapeutically relevant concentrations. The need for apoptosis verification through Annexin testing is highlighted, emphasizing the gap in comprehensively understanding the compound's effects on cell viability and death.
- 7. It is further emphasized that investigating MRC-5 cell lines using 3D assays would not only enrich the study's comprehensiveness but also potentially uncover differential responses between cancerous and non-cancerous cells in three-dimensional cultures
- 8. The paper does not provide a rationale for the selection of specific cell lines (MDA-MB-231, A549, DU145, and MRC-5) for the experiments.
-
-
Round 2
Reviewer 1 Report
Comments and Suggestions for Authors
accept
Reviewer 2 Report
Comments and Suggestions for Authors
The manuscript has been improved; however, the MTT results still lack an appropriate reference compound.
Round 3
Reviewer 2 Report
Comments and Suggestions for Authors
The manuscript has been improved.